# The Treatment Efficiency and Microbiota Analysis of *Sapindus mukorossi* Seed Oil on the Ligature-Induced Periodontitis Rat Model

**DOI:** 10.3390/ijms23158560

**Published:** 2022-08-02

**Authors:** Shih-Kai Lin, Yi-Fan Wu, Wei-Jen Chang, Sheng-Wei Feng, Haw-Ming Huang

**Affiliations:** 1School of Dentistry, College of Oral Medicine, Taipei Medical University, Taipei 11031, Taiwan; middlesky00@gmail.com (S.-K.L.); yfwu@tmu.edu.tw (Y.-F.W.); cweijen1@tmu.edu.tw (W.-J.C.); 2Department of Dentistry, Shuang Ho Hospital, Taipei Medical University, New Taipei City 235041, Taiwan

**Keywords:** *Sapindus mukorossi*, ligature-induced periodontitis rat model, micro CT, microbiota

## Abstract

Periodontitis is a common oral disease mainly caused by bacterial infection and inflammation of the gingiva. In the prevention or treatment of periodontitis, anti-bacterial agents are used to inhibit pathogen growth, despite increasing levels of bacterial resistance. *Sapindus mukorossi Gaertn* (SM) seed oil has proven anti-bacterial and anti-inflammation properties. However, the possibility of using this plant to prevent or treat periodontitis has not been reported previously. The aim of this study was to evaluate the effects of SM oil on experimental periodontitis in rats by using micro-CT and microbiota analysis. The distance between cementoenamel junction (CEJ) and alveolar bone crest (ABC) on the sagittal micro-CT slide showed that total bone loss (TBL) was significantly lower in CEJ-ABC distances between SM oil and SM oil-free groups on Day 14. Histology data also showed less alveolar bone resorption, a result consistent result with micro-CT imaging. The microbiota analyzed at phylum and class levels were compared between the SM oil and SM oil-free groups on Day 7 and Day 14. At the phylum level, *Proteobacteria*, *Firmicutes*, *Bacteroidetes*, and *Actinobacteria* were the dominant bacterium. *Firmicutes* in box plot analysis was significantly less in the SM oil group than in the SM oil-free group on Day 7. At the class level, *Bacteroidia*, *Gammaproteobacteria*, *Bacilli*, *Clostridia*, and *Erysipelotrichia* were the dominant bacteria. The bacteria composition proportion of *Bacilli*, *Clostridiay,* and *Erysipelotrichia* could be seen in the SM oil group significantly less than in t SM oil-free group on Day 7. Overall, the present results show that topical application of SM oil can reduce bone resorption and change bacteria composition in the ligature-induced periodontitis model. According to these results, it is reasonable to suggest SM oil as a potential material for preventing oral disease.

## 1. Introduction

Periodontitis is a common chronic oral inflammatory disease that leads to the destruction of tooth-supporting tissues, which may be caused by genetic variations [1], imbalanced oral microbiome [2], or elevated levels of systemic pro-inflammatory mediators in the blood [3,4]. The occurrence of periodontal pathogens with persistent inflammation from dental plaque destroys periodontal structures such as periodontal ligament, alveolar bone, cementum, and soft tissue. Notably, over 10% of adults suffer from severe periodontitis, making the condition a severe public health threat, with evidence from the 2019 Global Burden of Disease (GBD) study indicating that global population growth accounted for 67.9% of the increase in severe periodontitis [5,6,7]. Commonly used and effective treatment for periodontal disease are achieved through non-surgical approaches (i.e., scaling and root planning (SRP), sonic/ultrasonic treatment, locally delivered antibiotic drugs [8], diode laser treatment, photodynamic therapy) and surgical interventions such as guided tissue/bone regeneration (GTR/GBR), pocket reduction surgery, and osseous surgery (OS).

Natural plant-based products and traditional medicines have been considered promising candidates for the prevention and treatment of oral diseases. [9,10]. Sapindaceae (also called “wu-huan-zi” in Chinese), a natural plant-based product used in traditional Chinese medicine, has attracted increasing attention. *Sapindus mukorossi* (SM) Gaertn is one of the more commonly used major species. SM can be easily found in tropical and sub-tropical regions of the Pacific Rim [11]. Many studies have shown that the pericarp of the SM fruit also has medicinal value and positive biological effects, including insecticidal, antitrichomonal, anti-cancer, hepatoprotective, anxiolytic, molluscicidal, fungicidal, free radical scavenging, anti-inflammatory and anti-platelet aggregation activity [12,13]. More recent studies have also indicated that SM extract exhibits significant anti-microbial activities and could potentially be used as a source of agents to cure dental caries [14] and for skin care [15]. These studies have clearly shown that SM’s major anti-bacterial effect is from saponins extracted from the fruit’s pericarp [12,14,16].

Besides the pericarp, which makes up 56% of the SM fruit, SM contains a seed covered by a hard, black shell with kernel pulp inside [17,18]. The kernel of the SM seed contains an abundant amount of oil (23%), mainly composed of fatty acids and triglycerides [19]. While the inedible SM seeds have been regarded as a useless waste product in the past [20,21], possible dental applications of SM seed kernel have been recorded in the Compendium of Materia Medica (Mandarin: Bencao Gangmou), a Chinese pharmaceutical volume written almost 500 years ago. According to this work, the kernel of the SM seed can be used to treat gingival swelling and reduce oral malodor. Since it is well known that bacteria predispose the development of bad breath, the SM seed kernel is likely to provide anti-inflammatory and anti-bacterial effects, thus reducing periodontal disease and oral caries. An interesting modern in vivo experiment by Chen and colleagues concluded that SM oil could potentially promote the healing of skin wounds through anti-inflammatory and anti-microbial activities [13].

A report demonstrated that SM fruit pericarp extract could inhibit the growth of bacteria associated with oral diseases, including *P. gingivalis*, *A. odontolyticus*, *F. nucleatum* and *C. albicans* [22]. However, whether SM seed oil can be used to treat periodontal disease remains unknown. Since the bacterial communities that cause periodontitis are complex [23,24,25,26], clarifying the changes in the composition of microbiota associated with infected periodontal tissue will be the first step toward understanding the treatment effect of SM oil on periodontitis and to establish therapeutic strategies.

Accordingly, this study used a ligatured animal model of periodontal disease to determine correlations between microbiota and periodontitis, as well as to assess SM oil application through micro-CT and histological analysis during the progress of periodontal disease in rats. In addition, 16s rRNA sequencing techniques were used to gain deeper insights into inflammatory and microbiota changes in periodontitis after SM oil treatment.

## 2. Results

### 2.1. Micro-CT Analysis of Bone Resorption

The typical reconstructed micro-CT images and analysis of the bone resorption, such as CEJ-ABC distance and subsequent bony changes around CEJ, are shown in Figure 1. Compared to the blank control (Figure 1a), distances between CEJ and ABC in w/o oil groups on Day 7 (Figure 1b) and Day 14 (Figure 1c) were visibly increased. These results indicate that induction of periodontitis in the experimental animal model was effective. However, when the silk ligatures tied on the rat’s maxillary molar were pre-immersed with SM oil, the CEJ-ABC distance decreased on Day 7 (Figure 1d) and Day 14 (Figure 1e).

Figure 2 shows total bone loss (TBL) and furcation bone loss (FBL). These figures show that the change in furcation bone loss level showed no significant difference during the entire experimental period (*p* = 0.891 and 0.284 for Day 7 and Day 14, respectively) (Figure 2a). Nevertheless, a significant difference for TBL was seen in CEJ-ABC distances between sample teeth ligated with and without SM oil on Day 14 (*p* < 0.05) (Figure 2b).

### 2.2. Histologic Analysis of Periodontitis Progression

Figure 3 shows typical histopathologic images of the periodontitis progress of the experimental samples. Periodontal tissue in sample teeth tied with ligatures had obvious pouch-like periodontal pockets on Day 7 (Figure 3b) and Day 14 (Figure 3d). In addition, compared with the blank control group (Figure 3a), apical migration of the epithelium along the tooth surface was apparent on the exposed cementum when SM oil-free ligatures were used. When ligatures were pre-immersed with the SM oil, no visible change in inflammatory infiltration was found during the 14-day period (Figure 3c,e). However, we found that samples ligatured with SM oil-immersed silk had less alveolar bone resorption (Figure 3c,e) as compared with teeth ligatured with SM seed oil-free silk at both Day 7 (Figure 3b) and Day 14 (Figure 3d). In addition, unlike samples treated with SM oil-immersed ligature that showed thick and continuous epithelium, samples in the SM oil-free group exhibited thin and discontinuous epithelium (Figure 3f), and looser lamina propria and submucosa on Day 14 (Figure 3g).

### 2.3. Microbiota Investigation of Periodontitis Progression

In Figure 4, the overall heatmap distribution shows that microbial species become more diverse in the SM oil-free group at all time points. Compared to the oil-free group, sample teeth tied with SM oil-immersed ligatures had lower diversity at the family (Figure 4a) and genus (Figure 4b) levels.

Bacterial communities were compared between samples ligated with SM oil pre-immersed and SM oil-free silk ligature at phylum (Figure 5a) and class (Figure 5b) levels. As shown in Figure 5a, we found that *Proteobacteria*, *Firmicutes*, *Bacteroidetes*, and *Actinobacteria* were dominant both in SM oil (w/ oil) and SM oil-free groups (w/o oil). *Proteobacteria* was the most dominant bacterium in all samples. On Day 7, the relative abundance of *Proteobacteria* in the SM oil group was 50.41%, approximately 1.34 fold higher than the oil-free group (37.58%). After a 14-day experimental period, the proportions of the two groups became similar (Figure 6b). *Firmicutes* had significantly less relative abundance in the SM oil group (32.16%) compared to the oil-free group (45.68%) on Day 7 (Figure 6a). As with *Proteobacteria*, the proportion of *Firmicutes* detected in the two groups became similar on Day 14 (Figure 6a). The relative abundance of *Bacteroides* in samples ligated with SM oil pre-immersed ligature was 1.79%, slightly lower than in the SM oil-free ligature samples (2.26%) on Day 7. However, this result reversed on Day 14 (Figure 6a). The relative abundance of *Bacteroides* in SM oil and SM oil-free groups were 4.73% and 2.82%, respectively. The relative abundance of *Actinomycetes* detected in sample teeth tied with SM oil and oil-free samples were 15.41% and 14.42% on Day 7, respectively. These values increased to 32.95% (SM oil group) and 31.50% (SM oil-free group) on Day 14 (Figure 6d).

At the class level, *Bacteroidia*, *Gammaproteobacteria*, *Bacilli*, *Clostridia*, and *Erysipelotrichia* were the dominant classes found in the samples. *Gammaproteobacteria* was the dominant bacterium in both groups (Figure 5b). The proportion of *Bacilli* bacteria on Day 7 in the SM oil group (w/ group) was 25.72%, lower than samples treated with SM oil-free ligatures (w/o group) (33.93%). These values became similar by the end of the 14-day experimental period (Figure 7a). A similar tendency was found for *Clostridiay* (Figure 7b). During the entire experimental period, the relative abundance of *Erysipelotrichia* for the SM oil-free group was higher than in samples tied with SM oil ligature (Figure 7c). Interestingly, on Day 14, the value for SM oil-free group (1.87%) was almost 4.2 fold higher than in samples tied with SM oil pre-immersed silk ligatures (0.45%) (Figure 7c). On Day 7 the proportion of *Bacterodia* in the SM oil group was 1.78%, lower than in the SM oil-free samples (2.26%). This tendency had reversed by Day 14 (Figure 7d). For *Gammaproteobacteria*, the composition proportion for the SM oil group was higher than the SM oil-free group on Day 7 and Day 14 (Figure 7e).

### 2.4. Firmicutes/Bacteroidetes (F/B) Ratio

F/B ratio was considered to assess alternating inflammatory phases when SM oil pre-immersed ligatures were used to produce periodontitis. Figure 8 shows that on Day 7 the acute inflammatory phase for sample teeth treated with SM seed oil-free ligatures was obviously lower than in samples ligated with SM oil-free silk ligatures. However, the difference in the F/B ratio had fallen sharply by the end of the experimental period (Figure 8).

## 3. Discussion

Ligature-induced periodontitis was used in the present study to evaluate the treatment efficiency of SM oil in an animal model. This model is one of the most widely used methods to assess the relationship between periodontal disease and potential treatment materials [27,28]. Our micro-CT results (Figure 1 and Figure 2) showed that total bone loss (TBL) was significantly lower in the seed oil group than the oil-free group on Day 14 (Figure 2b). SM oil can reduce bone resorption in a ligatured animal model and maintain a healthy periodontal environment, suggesting that SM oil is a potential material for inhibiting the progression of periodontitis. Although a previous study by Goncalves-Zillo et al. found that Bdkrb1-/- mice with ligatured-induced periodontitis had increased bone loss in the furcation area [28], there is no significant difference in bone resorption found between SM oil-treated and oil-free samples in the root furcation area (Figure 2a) in this study. This phenomenon is due to the SM oil applied at the CEJ surface not reaching the furcation area. 

Typically, burst-destruction bone loss occurs within two weeks [29,30], which was confirmed by our histological experiments in which pouch-like surrounding bone was observed on Day 7 and Day 14 (Figure 3). However, the inflammatory response in ligated teeth was not apparent during the experimental period, possibly due to the sulcular epithelium of tested rats being keratinized (Figure 3f,g) [31]. From Figure 3g, thicker keratinized epithelium and dense lamina propria of the SM-oil treated sample were found when compared to the oil-free group (Figure 3f). Since a thickened keratinized epithelium acts as a barrier that provides more efficient resistance to bacterial invasion, it is reasonable that there is no obvious inflammatory response found in our experimental model [32,33]. This provides histological evidence to support SM oil’s action against periodontitis by maintaining the healthy status of surrounding soft tissue. 

The positive effect of SM oil on bone growth was first reported by Shiu et al. (2020) [34], which showed that SM oil exhibits a significant effect on dental pulp stem cells via alkaline phosphatase gene expression and extracellular matrix vesicle secretion, identifying the possible active component of SM oil for maintaining bone quality as β-sitosterol found in seed kernel pulp. In 2019, Yildirim et al. reported that β-sitosterol contained in morus nigra (known as the black mulberry) inhibited regional alveolar bone resorption in a rat periodontitis model via a reduction in MMP-8 expression [35]. MMP-8 has been reported as a collagenase that can destroy the extracellular matrix [36] in the soft tissue around teeth, which results in periodontitis [37,38]. In 2019, Mahmoudi et al. reported that a plant seed oil (*Alyssum homolocarpum*) promoted stem cell proliferation and differentiation through β-sitosterol inside the oil [39]. It was reported that SM oil also contains abundant β-sitosterol in the seed kernel pulp [13]. Since β-sitosterol exhibits various pharmacological and biological activities during bone regeneration [40], it is reasonable to speculate that β-sitosterol in SM oil plays an important role in preventing bone loss during the progression of periodontitis. 

With bacteria known to be the major factor inducing bone loss in periodontitis [28,30] and the ability to accumulating commensal microbial biomass to trigger a switch from homeostasis to inflammation [41], SM oil’s anti-bacterial properties suggest the material as a potential periodontitis inhibitor. Increased microbial diversity is characteristic of disease development in the root apex, and the increased specific bacterial communities are closely associated with disease progression [42]. The heatmap in Figure 4 shows that samples tied with SM oil-immersed ligatures exhibited lower diversity at both family and genus levels (Figure 4). These results provide another piece of evidence to strengthen the hypothesis that SM oil exhibits anti-periodontitis qualities. 

According to a study by Abe et al. [30], bacteria constitution is the major factor in the induction of bone loss in ligature-induced periodontitis. As shown in Figure 5, *Proteobacteria*, *Firmicutes*, *Bacteroidetes*, and *Actinobacteria* were dominant phyla in both ligated molars tied with oil-immersed and oil-free ligatures. This population is similar to microbiota found in the analysis of saliva, pulp chamber, and root apex [42] and is often identified in healthy subjects [43]. However, our results demonstrated that the microbial community of ligated molars changed after SM oil treatment. This result is consistent with previous findings that also showed a similar change in microbiota results among healthy and diseased periodontal tissues [44,45,46,47]. Previous reports have also indicated that *Proteobacteria* and *Actinobacteria* decreased in abundance while *Bacteroidetes* and *Firmicutes* increased during oral disease [48,49]. Although SM oil seemed not to influence *Actinobacteria* in our animal study (Figure 6d), we found that *Firmicutes* and *Bacteroidetes* had decreased in the SM oil group on Day 7 (Figure 6a,b). Our results also show that the average abundance of *Proteobacteria*, the most prevalent species in the blood [50,51] and gut [52], was higher in the SM oil-treated sample than in oil-free teeth on Day 7 (Figure 6c). Since *Proteobacteria* have been reported to be more abundant in healthy human sub-gingiva [24], this can be considered evidence that SM oil exhibits a positive effect on maintaining healthy bacteria populations at phylum level. 

At the class level, *Bacilli*, *Clostridia*, and *Erysipelotrichia* were inhibited by SM oil on Day 7 (Figure 7a–c), and *Erysipelotrichia* remained lower in the SM group on Day 14. These reduced bacterial populations would inhibit alveolar bone resorption [53] and periodontitis [54,55,56]. At the same time, SM oil’s positive effect on *Gammaproteobacteria* (Figure 7e) was more pronounced, confirming that SM oil tends to maintain the health of silk-ligated molars [49]. These results suggest that SM oil exhibits a positive effect on tooth health by maintaining a normal microbial community at the class level.

F/B ratio identified in the inflammatory phase is a good indicator of microbiota dysbiosis in the oral cavity [57,58]. In our study, the F/B ratio was lower in the SM oil group compared to the oil-free group (Figure 8) due to SM oil’s inhibition of *Firmicutes* overgrowth, also indicating that SM oil reduces the burst process of the inflammatory condition [59]. Our present results thus prove the beneficial effect of SM oil in preventing ligature-induced periodontitis and improving periodontal tissue health at an early stage.

The advantages of using SM oil over conventional agents as a treatment for periodontitis are that the product is a safe, natural, and sustainable. However, this study has several limitations. First, SM oil was applied only once at the initial stage. Because not providing a continuous application of the oil prevents observations of long-term effects, future studies could evaluate the effect of SM oil on chronic periodontitis across a longer timeframe. In addition, the main etiological agents of periodontal disease are different between humans and animals. For example, *Porphyromonas gingivalis*, *Aggregatibacter actinomycetemcomitans*, *Prevotella intermedia* and *Tannerella forsythia* play an important role in periodontal disease in humans. However, according to previous reports, these bacteria were only rarely detected in animals with periodontitis [60]. Because the periodontal microflora of the rat is still unclear, and because over 50% of common oral bacterial species have not been formally named [61], microbiota analysis at the species level was difficult in this experiment. Since the major impetus of this study was to evaluate the effect of SM oil on periodontitis, and because Qian et al.’s results indicate that microbiota analysis at the phylum level represent the health status and treatment efficiency in apical periodontitis [24], though species-level bacterial populations in periodontitis are well known in humans, microbiota analysis was not performed at this level in the current animal study.

## 4. Materials and Methods

### 4.1. Preparation of SM Oil

SM used in the experiment was obtained from a company (He-He Com. Ltd., Taipei, Taiwan). After removing fruit pulp and pericarp, seeds were washed under running water and then sterilized in distilled water before oil extraction. After drying in a cabinet drier (40 °C for 72 h), the seeds were milled to separate kernels. The cold-press extraction method was used to extract seed oil following the procedure outlined by Chhetri et al. [11]. Finally, the extracted oil was filtrated and stored in a freezer at −20 °C.

### 4.2. Animal Preparation and Experimental Design

The animal study procedure in this experiment was reviewed and approved by the Institution Animal Care and Use Committee or Panel (IACUC Approval No. MI202001-01). All efforts were made to minimize the number of animals used and reduce any suffering while producing reliable scientific data. A total of 15 pathogen-free eight-week-old SD rats were randomly and evenly assigned to two experimental groups and one blank control (five rats for each group). Rats in the two experimental groups were sacrificed on Day 7 and Day 14, respectively. The blank control was sacrificed on Day 0. In order to ensure a randomized experimental design, different operators grouped the SD rats and conducted the ligature experiment. During the experimental period, all animals were fed a standard diet and kept independently in clean cages with adequate ventilation. The experimental design is shown in Figure 9a.

### 4.3. Ligature-Induced Periodontitis Model

According to previous studies, the ligature method was used to prepare a periodontitis model [19,20,21,22]. Under pentobarbital anesthesia, 3-0 silk ligatures were tied on the rats’ bilateral maxillary second molar areas. In the experimental group, ten rats had ligatures pretreated with an immersion of SM oil (w/ oil) tied on their right side, while ligatures free of SM oil (w/o oil) were tied on the left side to serve as a comparison group (w/o oil) (Figure 9b). Five rats without ligatures served as the blank control group and were sacrificed on Day 0 [62]. To ensure the experiment was performed under blind conditions, different operators prepared ligature samples and ligatured the teeth of SD rats. Control group samples (w/o oil) were only immersed in normal saline.

Silk ligatures are predicted to induce periodontal inflammation with pathological biofilms forming both on ligatures and tooth surfaces [30,62]. According to a previous protocol, maxillary samples were collected at three time-points: Day 0, Day 7, and Day 14 [63]. Animals were sacrificed under anesthesia, and their jawbones were collected at the time or observation. After removing excessive soft tissue, preserved maxillary bones from the experimental groups, including samples ligated both with oil pre-immersed and oil-free silk, were split in half and collected in separate 1.5 mL test tubes, followed by fixation and storage in 10% formalin. The periodontal status condition of these maxillary bone samples was evaluated by micro-CT and histological images. Collected ligatures were preserved in a freezer at −20 °C prior to 16s rRNA sequencing.

### 4.4. Micro-CT Analysis

Maxillary samples were collected on Day 0, Day 7, and Day 14 after surgery and scanned with a SkyScan 1076 scanner (SkyScan, Bruker, Kontich, Belgium) with a resolution of 35µm at 90 kV. 3D images of the buccal and palatal sides were reconstructed using NRecon (version 1.7.4.2, Bruker). For bone loss evaluation, distances between the cemento-enamel junction (CEJ) and alveolar bone crest (ABC) were analyzed using CTAn (version 1.19.4.0, Bruker). Six sites, including BM (buccal-mesial), BF (buccal-furcation), BD (buccal-distal), PM (palatal-mesial), PF (palatal-furcation), and PD (palatal-distal) from each maxillary molar were chosen for measurement, and average distances were calculated. In this study, total bone loss (TBL) was defined as the sum of measured data from six sites (BM + BF + BD + PM + PF + PD). Furcation bone loss (FBL) was defined as the sum of measured values at BF and PF.

### 4.5. Histological Analysis

In order to assess the samples’ periodontal conditions, maxillae were dissected at 0, 7, and 14 days and subsequently fixed with 10% formaldehyde. Demineralization was performed according to Ayukawa et al. (1998) [64]. Briefly, both sides of each maxilla were decalcified with 10% ethylene-diamine-tetra-acetic acid solution (pH 7.0) (Ajax Finechem, Thermo Fisher Scientific; Taren Point, Australia) [65] for 4 weeks at 4 °C. Then tissue blocks were embedded in paraffin and cut into serial mesial-distal sections (5 μm thick) using an ultramicrotome (Bright 5040, Bright Instrument, Cambs, UK). After staining with hematoxylin and eosin (H&E) (HD Scientific Supplies; Wetherill Park, Australia) [65], histological images were acquired using a microscope slide scanner (OPTIKA, Ponteranica, Italia) [63].

### 4.6. 16s rRNA Extraction, Sequencing, and Bioinformatics Analysis

For sequencing and bioinformatics analysis, ligature samples (n = 3) from Day 7 and Day 14 were selected. Biofilm was extracted from ligatures and plaque on the tooth surfaces at the CEJ and furcation areas, and 16s genes on the ligatures were amplified by specific primers 341F-805R [5′-CCTACGGGNGGCWGCAG-3′ and 5′-GACTACHVGGGTATCTAATCC-3′]. All polymerase chain reactions (PCR) were performed in 25 μL reagents with 0.5 μL KAPA^®^ High-Fidelity PCR Master Mix (KAPA BIOSYSTEMS, Cape Town, South Africa). The final regent amount was 0.5 μM of forward and reverse primers and 1 ng DNA template. Amplification was performed with 30 cycles of denaturation at 95 °C for 30 s, annealing at 57 °C for 30 s, and elongation at 72 °C for 30 s, followed by 5 min extension at 72 °C. The amplified gene samples were purified using a commercial extraction kit (QIAquick Gel Extraction Kit, QIAGEN, Germantown, MD, USA). Purified gene sequences were processed using V3-V4 genomics analysis (Illumina MiSeq platform, Genomics BioSci & Tech Co., New Taipei City, Taiwan). The standard taxonomic identification procedure is length > 150 bp, with a default error rate = 0.1% with minimal overlapping > 10 bp. Genes with a similarity value of more than 97% were classified as the same operation taxonomic unit (OTU) identity using an open-source software platform (Mothur) with SILVA database (Microbial Genomics and Bioinformatics Research Group, Bremen, Germany) [66,67,68,69,70]. The gathered data was analyzed using open-source microbial community analysis software (QIIME, Quantitative Insights Into Microbial Ecology, ver. 1.9.0, Knight and Caporaso labs, http://qiime.org/1.4.0/; assessed on 15 November 2020) to provide microbiome information from raw DNA sequencing data [71]. Sequencing reads without ambiguous reads (n = 3) and chimeric sequences were processed for the bioinformatics analysis as previously described. Differences in bacterial community patterns among samples ligated with and without SM oil for Day 7 and Day 14 were evaluated using median and percentage.

### 4.7. Statistical Analyses

Differences in bone loss between groups were evaluated using t-test analysis. Differences at *p* < 0.05 were considered significant. Statistical analyses were performed using commercial software (SPSS Inc., Chicago, IL, USA). The groups’ differences in bacteria composition were evaluated using box plot analysis with commercial software (Microsoft Excel, Roselle, IL, USA).

## 5. Conclusions

This study demonstrates that SM oil can significantly decrease total bone loss (TBL) and bone resorption in a ligatured animal model. Our results also show that SM oil inhibits pathogenic bacteria associated with oral disease and suppress the F/B ratio. Therefore, SM oil can potentially protect periodontal tissue by altering the microbiota composition in the initial phase of periodontitis. The data shown in this study can be an important reference for future studies that examine larger sample sizes and in clinical studies in humans.

## Figures and Tables

**Figure 1 ijms-23-08560-f001:**
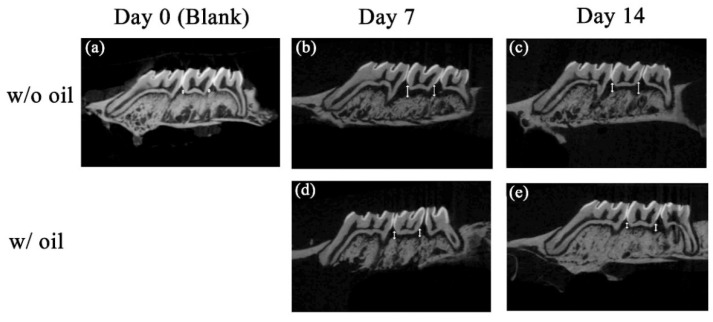
Micro CT images showing bone loss measurements for each group at Day 0 (**a**), Day 7 (**b**,**d**) and Day 14 (**c**,**e**). On the sagittal side, radiographic bone loss was detected from CEJ to ABC (white arrow). The bone loss for SM oil samples (**b**,**c**) were lower than the oil-free groups (**d**,**e**).

**Figure 2 ijms-23-08560-f002:**
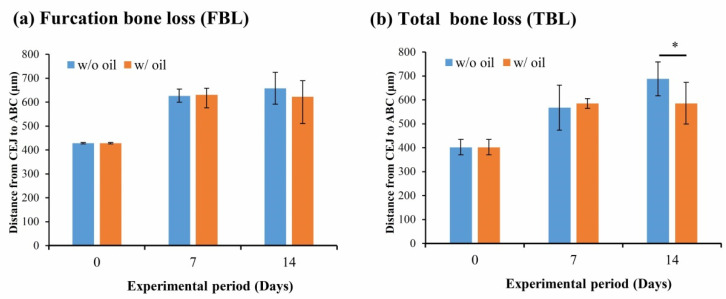
Influence of SM oil on alveolar bone loss at (**a**) furcation area and (**b**) total surrounding bone area. Quantitative analyses of bone loss were obtained by measuring CEJ-ABC distance w/ oil and w/o oil represented tooth samples ligated with or without SM oil pre-immersion, respectively. Significant differences between w/ and w/o oil groups were observed on Day14. * denotes *p* < 0.05.

**Figure 3 ijms-23-08560-f003:**
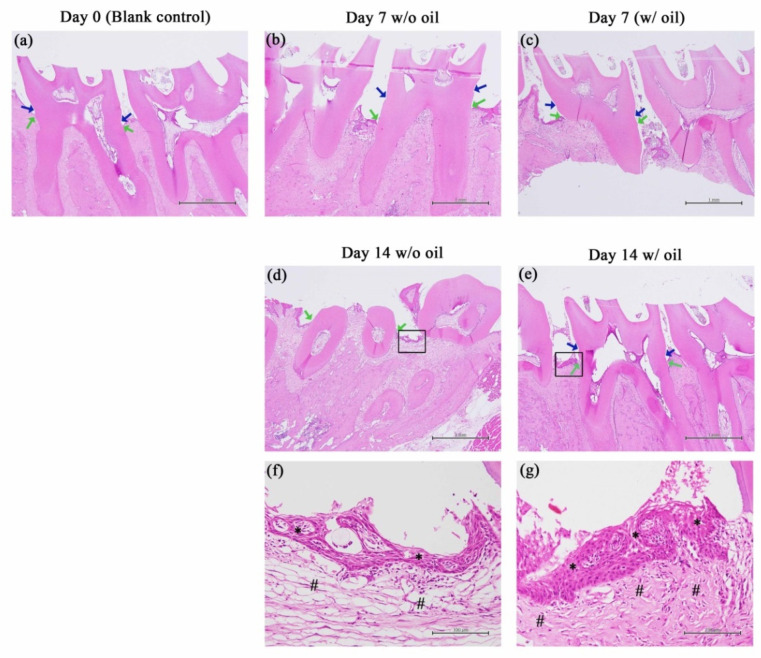
Histological results of tested molars at Day 0 (**a**), Day 7 (**b**,**c**) and Day 14 (**d**,**e**) with a magnification of 40×. Images of bone-surrounding tissue interface (black box) of oil-treated (**d**) and oil-free (**e**) samples on Day14 were enhanced by higher magnification micrographs of 400× (**f**,**g**, respectively). Blue arrows indicate cemento-enamel junction (CEJ). Green arrows indicate the most coronal level of the alveolar bone crest (ABC). * and # signs identify the epithelium and lamina propria, respectively. Scale bar is 1 mm for (**a**–**c**), and 100 μm for (**f**,**g**).

**Figure 4 ijms-23-08560-f004:**
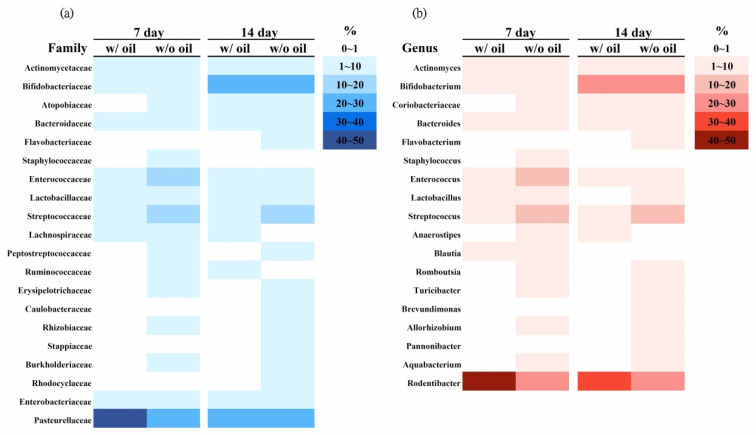
Heatmaps from microbiota analysis of samples treated with or without SM oil at (**a**) family and (**b**) genus levels during the experimental period. Values are presented in percentages.

**Figure 5 ijms-23-08560-f005:**
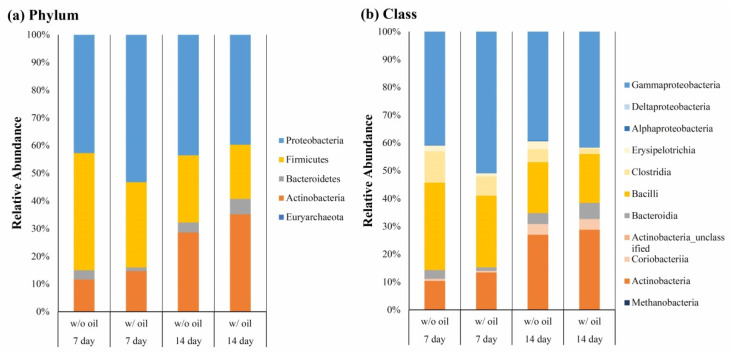
Relative abundance of bacteria in (**a**) phylum and (**b**) class levels on Day 7 and Day 14, when the tested teeth were treated with and without SM oil.

**Figure 6 ijms-23-08560-f006:**
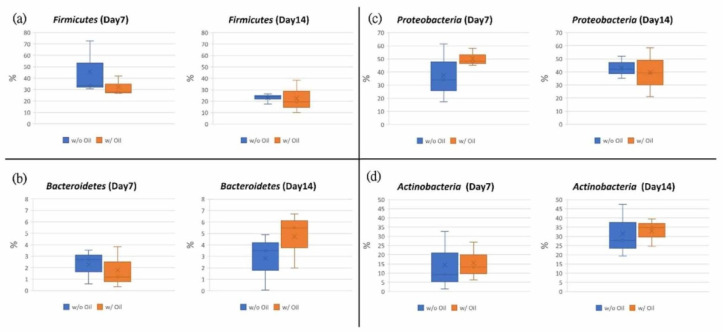
Tested teeth were subjected to ligature-induced periodontitis. Data from samples treated with and without SM oil were compared. Sub-figures show box plot analysis in the phylum of (**a**) *Firmicutes*, (**b**) *Bacteroidetes*, (**c**) *Proteobacteria* and (**d**) *Actinobacteria*.

**Figure 7 ijms-23-08560-f007:**
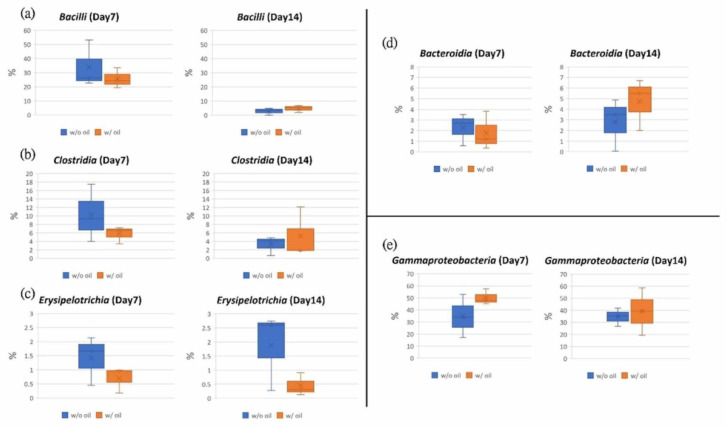
Tested teeth were subjected to ligature-induced periodontitis. Data from samples treated with and without SM oil were compared. Sub-figures showed the box plot analysis in the class of (**a**) *Bacilli*, (**b**) *Clostridia*, (**c**) *Erysipelotrichia*, (**d**) *Bacteroidia* and (**e**) *Gammaproteobacteria*.

**Figure 8 ijms-23-08560-f008:**
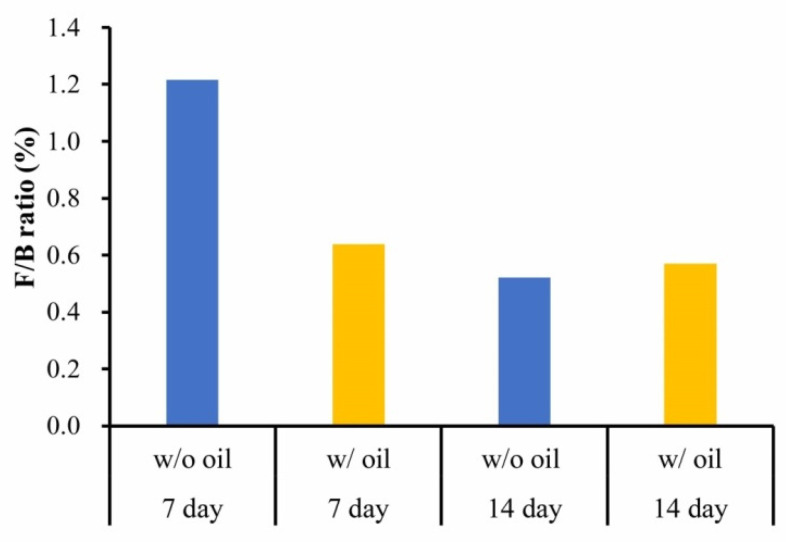
*Firmicutes*/*Bacteroidetes* (F/B) ratio from the median on Day 7 and Day 14 between samples treated with and without SM oil.

**Figure 9 ijms-23-08560-f009:**
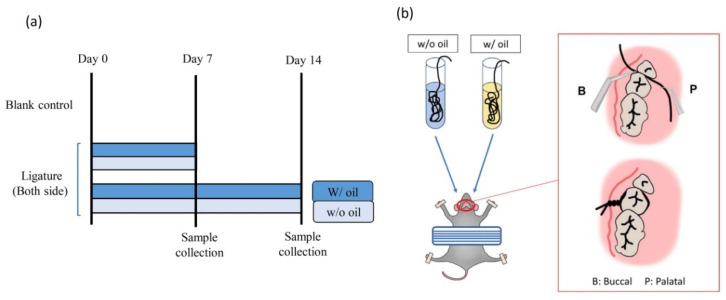
(**a**) Ligature-induced periodontitis experimental design with sacrifice time points. Blank control was the ligature-free samples at Day 0. In the ligature group, the second left molar was ligated with silk pre-immersed with SM oil (w/ oil). The corresponding tooth on the right side was tied with silk-free of SM oil (w/o oil) on Day 0. Rats were euthanized after 7 and 14 days. (**b**) Detailed procedures and preparation of the periodontitis animals were ligated with 3-0 silk around the cervical of the second molar of maxillary bone.

## Data Availability

Not applicable.

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
