# Peer review of "The Treatment Efficiency and Microbiota Analysis of Sapindus mukorossi Seed Oil on the Ligature-Induced Periodontitis Rat Model"

_ijms, 2022, doi:10.3390/ijms23158560_

Round 1
Reviewer 1 Report
Investigation the possibility of treating and preventing periodontitis with herbal preparations is actual task of the study.
However, there are some points that the authors should pay more attention:
1. "Periodontitis is a common oral disease caused by oral microflora dysbiosis" is an incorrect expression, since several factors may influence the pathogenesis of periodontitis. The microbial factor is the main, but not associated only with the dysbiotic state of the oral cavity.
2. A rat’s model of periodontitis was used. Is the periodontal microflora of the rat and human similar? Why were identified microorganisms only at the phylum and class level because the main periodontopathogenic microorganisms (genera, species) are well known (Tannerella forsythia, Prevotella, Porphyromonas et al).
3. "Effective methods of periodontitis prevention and treatment use anti-bacterial 53 agents such as tetracycline and penicillin to inhibit pathogen growth, despite rising 54 levels of bacterial resistance [4]" - currently, in addition to AB, a fairly wide arsenal of non-drug means of sanitation is used in dentistry (ultrasound, laser, etc.).
4. micro-CT in the abstract without decoding
5. Primers should be given in materials and methods
6.The authors could refer to some more most recent references (2020-2022).
Author Response
Reviewer 1:
* Comment 1.1: "Periodontitis is a common oral disease caused by oral microflora dysbiosis" is an incorrect expression, since several factors may influence the pathogenesis of periodontitis. The microbial factor is the main, but not associated only with the dysbiotic state of the oral cavity.?
Response: We sincerely thank the reviewer for pointing out these mistakes. We have added more descriptions to highlight this information and added recent references on the first paragraph of Introduction as below:
“Periodontitis is a common chronic oral inflammatory disease that leads to the de-struction of tooth-supporting tissues, which may cause by genetic variations [1], imbalance of oral microbiome [2], and elevated level of systemic pro-inflammatory mediators in the blood [3, 4]”
* Comment 1.2: A rat’s model of periodontitis was used. Is the periodontal microflora of the rat and human similar? Why were identified microorganisms only at the phylum and class level because the main periodontopathogenic microorganisms (genera, species) are well known (Tannerella forsythia, Prevotella, Porphyromonas et al).
Response: We thank the reviewer for pointing this out. In the revised manuscript, we added new paragraphs at the last parrgraph of Discussion to explain the method adopted in this study as following:
The main etiological agents of periodontal disease are different between humans and animals. For example, Porphyromonas gingivalis, Aggregatibacter actinomycetemcomitans, Prevotella intermedia and Tannerella forsythia play an important role in periodontal disease in humans. However, according to previous reports, these bacteria were almost not detected in dogs with periodontitis [Kato et al. (2011)]. Because the periodontal microflora of the rat is still unclear and over 50% of common oral bacterial species were not formally named [Dewhirst et al. (2010)], the microbiota analysis at the species level is difficult in this experiment. Since the major impetus of this study was to evaluate the effect of SM oil on periodontitis, and because Qian et al.’s results indicate that microbiota analysis at the phylum level represent the health status and treatment efficiency in apical periodontitis [Quin et al. 2019], though species-level bacterial populations in periodontitis are well known in humans, microbiota analysis was not performed at this level in the current animal study.
* Comment 1.3: "Effective methods of periodontitis prevention and treatment use anti-bacterial 53 agents such as tetracycline and penicillin to inhibit pathogen growth, despite rising 54 levels of bacterial resistance [4]" - currently, in addition to AB, a fairly wide arsenal of non-drug means of sanitation is used in dentistry (ultrasound, laser, etc.).
Response: We thank reviewer for the insightful comments. We have corrected these sentences in the revised manuscript as shown in the first paragraph of Introduction as below.
“The main effective treatment methods for periodontal disease can be achieved by non-surgical approaches (i.e., scaling and root planning (SRP), sonic/ultrasonic treatment, locally delivered drugs of antibiotics [8], diode laser, photodynamic therapy) and surgical intervention such as guided tissue/bone regeneration (GTR/GBR), pocket reduction surgery and osseous surgery (OS).”
* Comment 1.4: micro-CT in the abstract without decoding
Response: We thank the reviewer for pointing this out. The sentences have been revised in Abstract of the revised manuscript for better explanation as following:
“The distance of cementoenamel junction (CEJ) to alveolar bone crest (ABC) from the sagittal slide of micro-CT showed that total bone loss (TBL) was significantly lower in CEJ-ABC distances between the SM oil and SM oil-free group on Day 14. Histology data also showed less alveolar bone resorption, which had a consistent result with micro-CT”
* Comment 1.5: Primers should be given in materials and methods
Response: We thank for the reviewer’s suggestion. We have added more descriptions to highlight this information to the first paragraph of section 4.6 as below:
“Biofilm was extracted from ligatures and plaque on the tooth surfaces at the CEJ and furcation areas. 16s genes on the ligatures were amplified by specific primers 341F-805R [5’-CCTACGGGNGGCWGCAG-3’ and 5’-GACTACHVGGGTATCTAATCC-3’].”
* Comment 1.6: The authors could refer to some more most recent references (2020-2022).
Response: We thank for the reviewer’s suggestion. Several references from 2020 to 2022 were adopted to update the knowledge in the revised manuscript.
Reviewer 2 Report
Dear Authors,
I read the manuscript with interest. I think the research is well planned and carried out, and the manuscript is well written. I just have a few comments:
In figure 4 it is necessary to explain the meaning of the values included in the legend (scale)
In statistical analysis, p-values should be provided even if a non-significant association was detected
Statistical testing for bacteria composition should be carried out and reported. Comparing the percentages is not enough to assess the significance of the observed differences.
Best regards
Author Response
Reviewer 2:
I read the manuscript with interest. I think the research is well planned and carried out, and the manuscript is well written. I just have a few comments:
Author Response: We really thank for the reviewer’s positive feedback on our manuscript.
* Comment 3.1: In figure 4 it is necessary to explain the meaning of the values included in the legend (scale)
Author Response: We thank the reviewer to point out this typo error. The values in Figure 4 are in percentage. In the revised version, we added the percent signs to the figures and mentioned these in the legend.
* Comment 3.2: In statistical analysis, p-values should be provided even if a non-significant association was detected
Author Response: We thank the reviewer for this comment. P-values of non-significant association in Fig. 2 were added to the manuscript.
* Comment 3.3: Statistical testing for bacteria composition should be carried out and reported. Comparing the percentages is not enough to assess the significance of the observed differences.
Response: We thank the reviewer for this comment. The significance of the bacteria composition is indeed important for bacteria composition analysis at the species level. However, the species level analysis was not performed in this study due to the real oral bacteria in rat mouth is still unknown or un-named. Thus, refer to the previous study (Qin et al. in ref. 24), the microbiota analyses in phylum level and population diversity at family and genus levels were expressed in percentage bar chat and heat map, respectively.
Reviewer 3 Report
The aim of the manuscript titled „The treating efficiency and microbiota analysis of Sapindus 2 mukorossi seed oil on the ligature-induced periodontitis rat model” was to evaluate the effects of SM oil on experimental periodontitis in rats by using micro-CT and microbiota analysis. The results showed that topical application of SM oil can reduce bone resorption and change bacteria composition in the ligature-induced periodontitis model.
Overall, the manuscript is well executed, however some sections are rather long and hard to follow.
Introduction: The etiology of periodontitis should be explained more thoroughly. The authors should update the Reference list and underscore the role of microbiota, inflammatory rection and genetic of the host ( doi: 10.23736/S0026-4970.18.04198-5; doi: 10.1002/JLB.5RU1021-524R; doi: 10.3390/biom12040552). In my personal opinion paragraph 2 and paragraph 3 should be shortened. The aim of the manuscript is well defined.
Results: The results are clearly presented, and the additional CT images and histological pictures add to the value.
Discussion: This part seems a little bit chaotic and needs rearrangement. Some information are repeated. The limitations section should appper in the penultimate paragraph. I suggest adding also advntages of the study. The instruction and recommendations for future studies should also be added. The last paragraph should focus on main conclusions of the study.
Materials and methods: Was sample size calculated? What was the method used for randomisation? And what about blnding? Please explain and add to the appropriate parts of the manuscript.
Author Response
Reviewer 3:
Overall, the manuscript is well executed, however some sections are rather long and hard to follow.
Author Response: We thank this comment from the reviewer. In the revised manuscript, the length of Introduction and Discussion has been shortened and divided into several shorter parahraphs.
* Comment 2.1: Introduction: The etiology of periodontitis should be explained more thoroughly. The authors should update the Reference list and underscore the role of microbiota, inflammatory rection and genetic of the host ( doi: 10.23736/S0026-4970.18.04198-5; doi: 10.1002/JLB.5RU1021-524R; doi: 10.3390/biom12040552). In my personal opinion paragraph 2 and paragraph 3 should be shortened. The aim of the manuscript is well defined.
Author Response: We really appreciate the reviewer’s concerns and comments. In the revised manuscript, the texts in paragraph 2 and 3 in Introduction were reduced. In addition, we have added more descriptions to highlight this information and added recent references to the first paragraph of Introduction as below.
“Periodontitis is a common chronic oral inflammatory disease that leads to the de-struction of tooth-supporting tissues, which may cause by genetic variations [1], imbalance of oral microbiome [2], and elevated level of systemic pro-inflammatory mediators in the blood [3, 4]”
* Comment 2.2: The results are clearly presented, and the additional CT images and histological pictures add to the value.
Author Response: We really thank for the reviewer’s positive feedback on our manuscript.
* Comment 2.3: Discussion: This part seems a little bit chaotic and needs rearrangement. Some information are repeated. The limitations section should appear in the penultimate paragraph. I suggest also adding advantages of the study. The instruction and recommendations for future studies should also be added. The last paragraph should focus on main conclusions of the study.
Author Response: We thank for the reviewer’s suggestion. The discussion section has been rewritten for better explanation in the revised manuscript. The limitation of this study was moved to the last paragraph before conclusion. The advantages and future works were also added to the same paragraph followed by the description of limitation statemen. A new conclusion section has been moved to the end of the discussion paragraph.
* Comment 2.4: Was sample size calculated? What was the method used for randomization? And what about blinding? Please explain and add to the appropriate parts of the manuscript.
Author Response: We really appreciate the reviewer’s concerns and comments. The sentences about sample size have been added to the section 4.2 in the revised manuscript. In addition, The randomized method was added to section 4.2 and 4.3.
Round 2
Reviewer 1 Report
The authors have done a good job and have corrected the article. Introduction, materials\methods, discussion are sufficiently improved. The article may be recommended for publication.